# 4D Optical fibers based on shape-memory polymers

Clément Strutynski [1] ✉, Marianne Evrard[1], Frédéric Désévédavy[1], Grégory Gadret[1], Jean-Charles Jules[1], Claire-Hélène Brachais[1], Bertrand Kibler[1] & Frédéric Smektala [1]

Adaptative objects based on shape-memory materials are expected to significantly impact numerous technological sectors including optics and photonics. In this work, we demonstrate the manufacturing of shape-memory optical fibers from the thermal stretching of additively manufactured preforms. First, we show how standard commercially-available thermoplastics can be used to produce long continuously-structured microfilaments with shape-memory abilities. Shape recovery as well as programmability performances of such elongated objects are assessed. Next, we open the way for light-guiding multicomponent fiber architectures that are able to switch from temporary configurations back to user-defined programmed shapes. In particular, we show that distinct designs of fabricated optical fibers can maintain efficient light transmission upon completion of multiple temperature-triggered bending/straightening cycles. Such fibers are also programmed into more complex shapes including coils or near 180 ° curvatures for delivering laser light around obstacles. Finally, a shape-memory exposed-core fiber is employed in fiber evanescent wave spectroscopy experiments to optimize the performance of the sensing scheme. We strongly expect that such actuatable fibers with light-guiding abilities will trigger exciting progress of unprecedented smart devices in the areas of photonics, electronics, or robotics.

In recent years, both the variety and complexity of fiber profiles manufactured by the thermal drawing process have greatly diversified while the intricacy level of the tasks they are assigned has considerably widened[1–3]. Fibers are no longer passive objects, on the contrary, they can now combine a multitude of optical, electrical, and mechanical, functionalities providing them the ability to sense and react to their surroundings[4]. Fiber-based devices can even dynamically adapt to their environment and are able to perform three-dimensional motion by means of a tendon-driven mechanism to reach and navigate exiguous spaces[5]. Actuatable light-guiding structures have also been proposed using coating approaches of single-material microfibers, either taking advantage of the deposition of photothermally bendable hinges or photonic crystal periodic multilayers[6,7]. Taking into account

applications unrelated to optics, elongated tubular structures with shape-memory abilities fabricated through the preform-to-fiber drawing process have also proven useful for the production of conformable medical implants[8].

Such advances are in line with the recent and extraordinary progress made in the field of 4D printing[9,10]. This process involves the use of smart materials with additive manufacturing, thus creating objects capable of recovering a predefined shape when subjected to an external stimulus[11]. Shape-memory objects are therefore very adaptable, and they allow the development of innovative actuators at the nano-, micro-, and centimetric scales[12]. However, a major limitation of this technique is that the size of the fabricated devices is often limited relatively to the printing resolution (resolution to object dimensions

[1]Laboratoire Interdisciplinaire Carnot de Bourgogne (ICB) UMR 6303 CNRS-Université de Bourgogne, 21078 Dijon, France. ✉e-mail: clement.strutynski@u-bourgogne.fr

ratio around $10^3$–$10^5$). By contrast, thermal drawing is a processing platform that is not concerned with this constraint, thus fiber geometries structured at the micrometric scale and of several hundred or even thousands of meters in length can be produced. Bringing together 4D printing and fiber drawing is therefore highly desirable and could help solve the challenges of complexification, integration, and miniaturization of next-generation components for photonics, electronics, or robotics, as well as address low-cost and high-volume manufacturing problematics. The fabrication of preforms via 3D printing techniques and their subsequent stretching has proven to be useful for producing ever more sophisticated fiber architectures with unprecedented functionalities[13–15]. Air/polymer structures were initially developed due to their simplicity but with the improvement of additive manufacturing technologies[16], more intricate architectures could be implemented including hollow-core profiles or non-cylindrical geometries[17,18]. A lot of efforts have specifically been deployed for the fabrication of fibers with good optical quality using the technique, i.e. with defect-free structures and low attenuation levels[19]. The range of materials used has also considerably widened, now encompassing doped polymer matrices or even inorganic compounds such as soft glasses and the all-important silica[13,20,21]. However, although a single fiber can nowadays gather a multitude of different materials (glasses, metals, polymers, semiconductors, etc.) organized in intricate configurations[22,23], exploiting shape-memory (SM) materials for thermal stretching has never been reported to date. Integration of such materials within thin microfibers still holds unexplored potential to increase the design range of future waveguiding devices and develop new strategies for light collection or delivery[24], with applications in health, remote sensing, and robotics[25,26].

In this work, we take advantage of the highly scalable preform-to-fiber drawing process to produce tens of meters of continuous polymer-based optical fibers with shape-memory abilities. The preforms are additively manufactured from standard commercially-available thermoplastics. In particular, the elongated objects integrate semi-crystalline polylactic acid (PLA) domains which are responsible for the shape-memory effect. First, we report on fibers with arbitrarily structured multi-component cross-sectional geometries, thus featuring the elementary attributes of shape-memory objects, i.e. shape recovery and programmability. Next, light-guiding elongated structures are produced using the above-mentioned methodology and their transmission performances during the completion of bending/straightening shape recovery cycles from different curvatures are investigated. Finally, an adaptive sensor based on evanescent wave absorption spectroscopy (FEWS) is developed using an exposed-core shape-memory fiber. The ability of the waveguide to alter its shape is leveraged to dynamically optimize the performance of the device.

## Results

### Thermal drawing of shape-memory fibers

We describe here both the fabrication and characterization of a fiber that combines multiple polymers and exhibits shape-memory properties. The fiber is prepared by the preform-to-fiber drawing process as depicted in Fig. 1 . Polylactic acid (PLA) is chosen as the polymer with shape-memory properties[27]. PLA is well known for its shape-memory properties, which result from the presence of shape-switching domains within its structure that are able to perform elastic recoiling upon adequate stimulation[28]. However, PLA is a semi-crystalline polymer, and is therefore not suitable for fiber drawing, because the process requires a material that gradually softens upon heating. Indeed, semi-crystalline polymers, unlike amorphous thermoplastics, generally remain solid until they rapidly transform into a low-viscosity liquid when reaching temperatures above their melting point $T_m$. To overcome this limitation, semi-crystalline materials have to be associated, in most cases, with amorphous claddings (glasses or polymers)

to be successfully integrated into elongated structures[29–31]. For this reason, PLA is not a widely used polymer for fiber development through the preform-to-fiber method[32,33].

In the present work, polyethylene terephthalate glycol (PETG) is used as the amorphous cladding to control the flow of molten PLA during the thermal elongation process. The PLA and PETG parts (description in Fig. 1a, b respectively) are additively manufactured using an entry-level Fused Deposition Modeling (FDM) desktop printer (Ultimaker S3). Here, an H-shaped PLA core is inserted inside a hollow PETG cladding (see Fig. 1c) and the assembly is thermally elongated (Fig. 1d) using standard drawing equipment. Tens-of-meter-long continuous fibers are produced in this manner, as depicted in Fig. 1e, where ≈60 meters of shape-memory fiber drawn from a single preform are shown stored on a spool. Fig. 1f, g shows respectively a picture and a 2D diagram of the fiber cross-section.

Overall, the preform profile is well preserved, which proves that PLA and PETG can be successfully co-drawn to produce fibers with controlled geometries. In particular, PETG is selected as the cladding material for two reasons: (i) its drawing temperature is higher than the melting temperature of PLA, which is important for the drawing process, and (ii) its glass transition temperature is relatively close to PLA's, which facilitates the completion of shape recovery cycles of the fiber. The characteristic temperatures of PETG and PLA are extracted from the DSC curves plotted in Fig. 1h and recapped in Table 1. Next, we evaluate the shape-memory performances of fabricated fibers. By common acknowledgement, the practical strategies to control shape-memory effects in thermoplastics is to take advantage either of the glass transition temperature $T_g$ or of the melting temperature $T_m$ of the ordered phases[34]. At the macromolecular scale, the shape-memory behavior is attributed to the presence of netpoints and switchable segments in the polymer structure[28]. Netpoints are rigid domains that can be microscopic phases, crosslinking points or entanglements and are responsible for permanent shape fixation[35]. Switchable segments, on the other hand, are deformable domains, such as the amorphous phase. In PLA, above $T_m$, the crystalline phases melt, and chain mobility increases, allowing the polymer to be easily shaped. The programming stage consists in cooling down the material while applying a strain, i.e. holding the polymer in a user-defined shape. During this procedure, netpoints and switchable segments form which fixates the programmed configuration. At temperatures close to $T_g$, netpoints remain in place while switchable segments possess enough chain mobility to be altered (stretched, bent, etc.), enabling the polymer to be deformed into temporary configurations. However, mechanical stress is stored in these domains during the process. When the material is heated above $T_g$, the switchable domains gain mobility and release the stored energy by recoiling, thereby recovering the permanent shape. PLA is heated above its melting temperature during the thermal elongation process and rapidly cools as it passes the neckdown region of the preform, thus solidifying. This procedure is equivalent to a programming stage for the shape-memory polymer. As the fiber is pulled vertically on the drawing tower, the shape which is programmed after the thermal drawing is a straight line. First shape recovery tests are performed on the as-drawn fibers, as described in Fig. 2. Here, a ≈ 7-cm-long portion of PETG/PLA fiber is deformed at room temperature to obtain a temporary shape which is in that instance a square spiral (see top left picture in Fig. 2). For the PLA and PETG matrices, the strain is thus fixed under their $T_g$, meaning that residual chain mobility in the amorphous domains authorizes deformation of the materials even at room temperature. When the manually applied stress is then removed at room temperature, the PETG/PLA fiber holds its temporary shape as long as the $T_g$ of the PLA core and PETG cladding are not exceeded. The shape-memory fiber is then immersed in deionized water at ≈80 °C (above the $T_g$ values of PLA

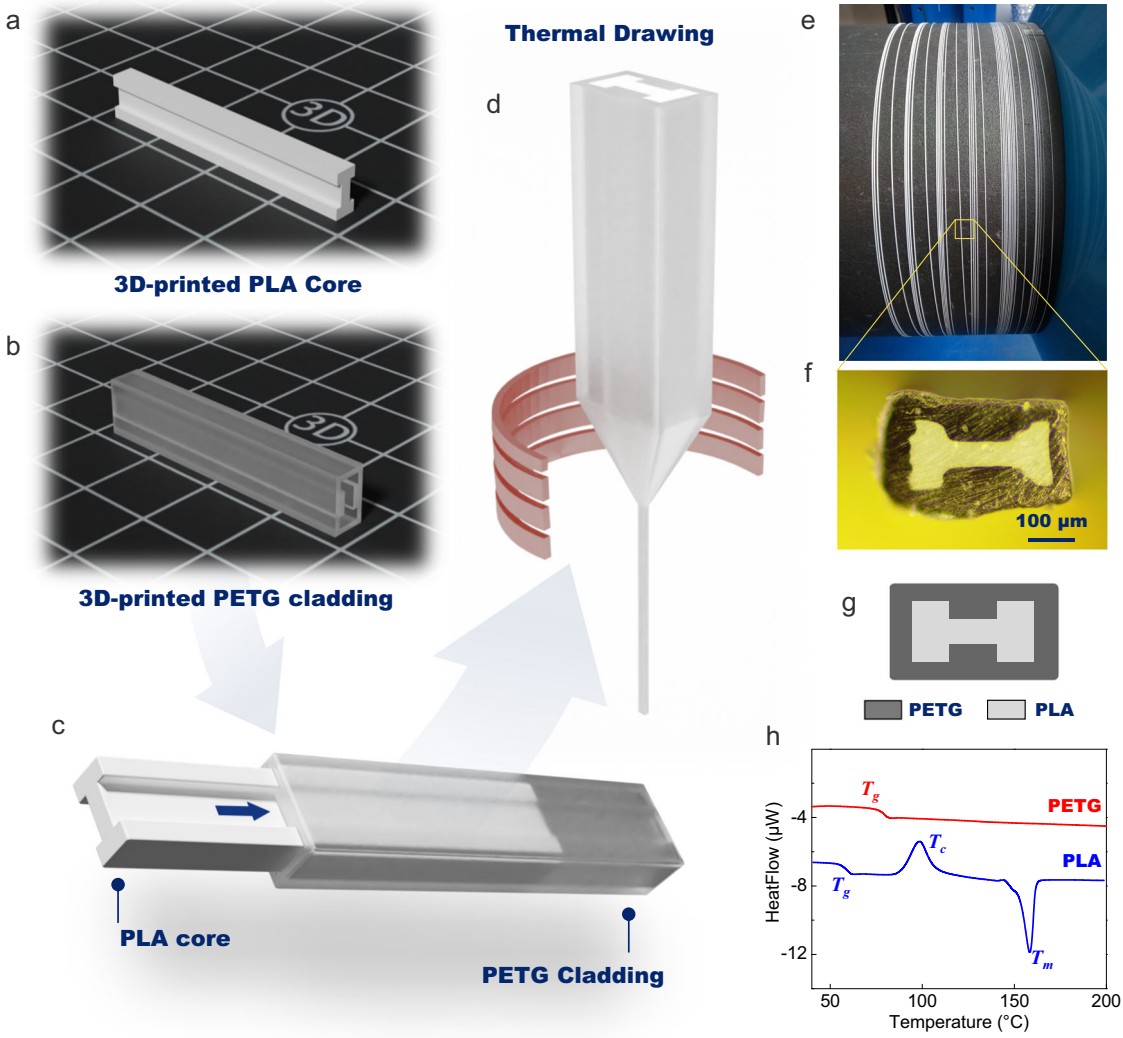

**Fig. 1 | Design and fabrication of shape-memory fibers.** 3D diagram of the **a** Polylactic acid (PLA) and **b** polyethylene terephthalate glycol (PETG) 3D printed parts used for the preform elaboration. **c** 3D diagram of the preform which is **d** thermally drawn into meters of shape-memory fiber. **e** Picture of ≈60 m of continuous shape-memory fiber manufactured from a single preform. **f** Cross-sectional view of the fabricated fiber and (**g**) 2D diagram of the cross-sectional geometry of the fiber. **h** DSC traces of PETG (red line) and PLA (blue line).

**Table 1 | Glass transition temperature $T_g$, temperature of onset crystallization $T_c$, melting temperature $T_m$, the refractive index at 640 nm $n_{640nm}$, Young's modulus, and elongation at break of selected thermoplastic polymers[37,38,42]**

| Polymer | Name | $T_g$ [± 3 °C] | $T_c$ [± 3 °C] | $T_m$ [± 3 °C] | $n_{640\ nm}$ | Young's modulus (MPa) | Elongation (%) |
|---|---|---|---|---|---|---|---|
| Polyethylene terephthalate glycol | PETG | 79 | n/a | n/a | 1.573 | 1000 | 240 |
| Polylactic acid | PLA | 59 | 87 | 150 | n/a | 3700–4100 | 4–6 |
| Polystyrene | PS | 90 | n/a | n/a | 1.587 | 3000–3500 | 2–3 |
| Acrylonitrile butadiene styrene | ABS | 105 | n/a | n/a | 1.522 | 1810–2390 | 8–20 |

n/a stands for non-applicable.

and PETG) without any applied stress, and is able to deploy into its original straight shape in a few tens of seconds (see bottom left picture in Fig. 2). It is worth mentioning that no material flowing is observed due to the physical crosslinking of PLA by crystalline parts and the highly viscous amorphous state of PETG. Going further, we now try to assess the programmability of the fabricated shape-memory fiber, i.e. its ability to retain and recover a particular shape. This consists in programming its PLA domains by heating PLA above its melting temperature $T_m$. In practice, the programming procedure of thin elongated objects is the following: (i) the fiber is

molded on a template object that determines the final shape, (ii) the assembly is heated above the melting temperature of PLA (here 1 min at 190 °C) to reprogram the semi-crystalline material (duration of the programming stage may vary depending on the fiber diameter), (iii) the fiber is rapidly cooled down to room temperature and subsequently released from the template object. A summary of the above programming procedure is depicted in Fig. 3a. As a first experiment, we programmed our H-shaped PLA core fiber into a coil. Remarkably, after deformation at room temperature, the fiber is able to get back to its programmed shape when immersed in a

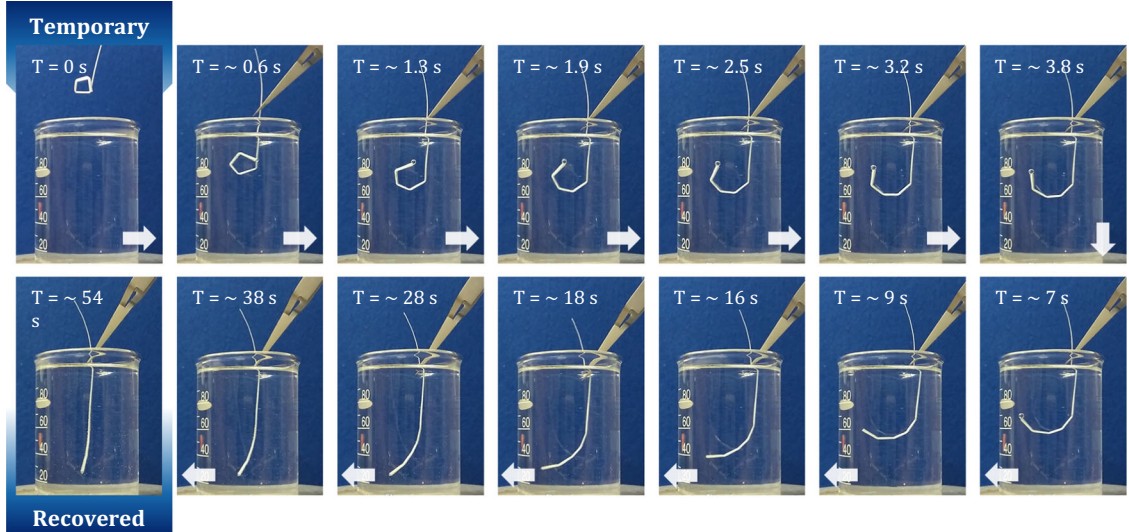

**Fig. 2 | Recovery cycle of an as-drawn shape-memory fiber.** Due to the nature of the thermal drawing process, the programmed shape of the as-drawn fiber is a straight line. The Polylactic acid (PLA)/polyethylene terephthalate glycol (PETG) fiber is deformed at room temperature into a temporary shape (square spiral) and subsequently immersed in deionized water at ≈80 °C. The shape-memory fiber deploys to its original straight shape in approximately 50 s.

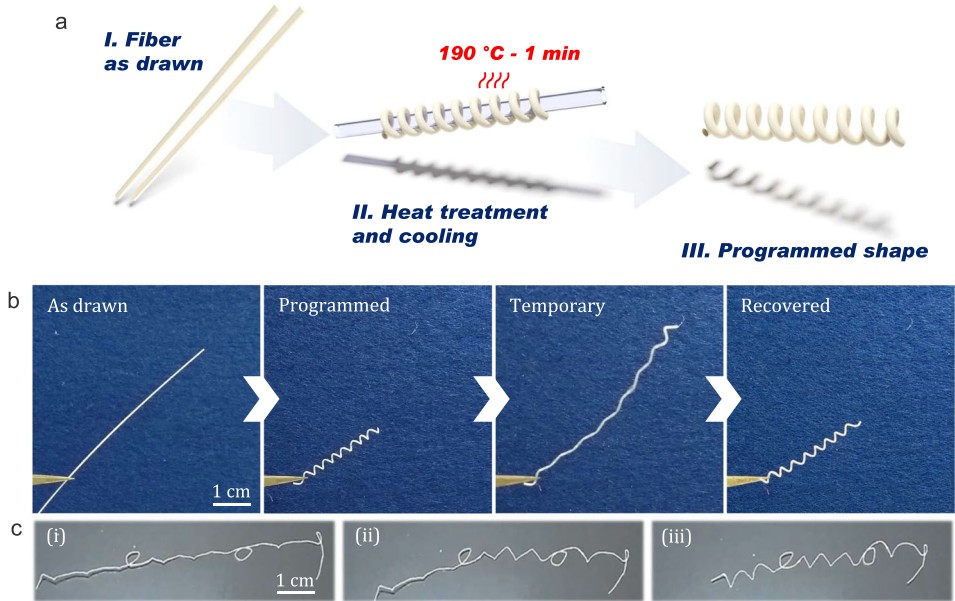

**Fig. 3 | Post-fiber drawing programming of shape-memory fibers. a** Description of the programming procedure of a shape-memory fiber: I- the as-drawn fiber has a straight shape, II- the fiber is molded on a template object and heated at 190 °C for 1 min and rapidly cooled down to room temperature, and III- 3D diagram of the programmed shape after the fiber is released from the template object. **b** Programming and shape recovery cycle of a shape-memory fiber programmed into a coil. **c** Shape-memory fiber (i) in a temporary configuration, (ii) in transition, and (iii) in its programmed shape spelling memory in cursive.

bath of deionized water at 80 °C. Fig. 3b shows pictures (from left to right) of the fiber as drawn, then programmed into a coil, deformed into a random temporary shape, and finally with the programmed form restored (after immersion in hot water). More complex shapes can be obtained from the SM fiber, as shown in Fig. 3c. Comparably to what is described above, the fiber is shaped into the word memory and held in position with a high-temperature adhesive tape. The sample is heated at 190 °C for 3 min, rapidly cooled down to room temperature, and finally released from its constrained position. The shape-memory fiber is deformed at room temperature (Fig. 3c-i) and recovers its programmed shape when heated on a hot

plate at ≈90 °C (Fig. 3c-ii) and finally forms the word memory again (Fig. 3c-iii).

## Light-guiding shape-memory fibers

After the basic features of shape-memory objects (namely, shape recovery and programmability) have been demonstrated on fibers fabricated by means of the thermal drawing process, we investigate more complex geometries in this section. In particular, the ability to produce light-guiding shape-memory elongated structures from the proposed methodology is assessed. Three different fiber architectures are proposed here. The first one is a hollow fiber or a fiber sleeve

(referred to as SM-sleeve) which can be combined with standard, commercially available fibers (silica, fluoride, chalcogenide or tellurite fibers, polymethyl methacrylate (PMMA) fibers, etc.). The two other structures are all-solid step-index geometries with either a polystyrene or a polyethylene terephthalate glycol core (respectively referred to as SM-PS and SM-PETG). The different materials are selected for both their rheological (for co-drawing) and optical (in terms of refractive index) compatibility. As described in the previous section, the polymer optical fibers (POFs) are prepared from the thermal stretching of additively manufactured preforms. They are built from a main PETG frame exhibiting (i) two lateral slots in which PLA parts are inserted and (ii) a central hole (Fig. 4a). The central hole is either kept empty (to produce the SM-sleeve structure) or filled with a polystyrene rod (to produce the SM-PS optical fiber) or filled with an ABS tube, acting as the optical cladding, in which a PETG core is inserted (to produce the SM-PETG optical fiber). Note that the central piece used for the core is not 3D-printed, which helps in minimizing optical attenuation although acceptable optical losses of 0.4 dB/cm can be achieved in 3D-printed polymer fibers[19]. A 2D diagram and pictures of the SM-sleeve, SM-PS, and SM-PETG architectures are given in Fig. 4b–d, Fig. 4e–g and Fig. 4h–j respectively. Overall, the profile of the starting preform is well transferred to the final fibers and the homothety is respected. It is demonstrated that meters of fibers with preserved cross-sectional profiles are produced from the methodology proposed here. The fiber profile is designed so that PLA occupies a significant part of the total volume of the overall structure (≈30 %). The quasi-cylindrical symmetry of the PLA inclusion also helps the final fiber to be steerable equally in any azimuthal direction.

As a first experiment, shape-memory performances of the SM-sleeve structure are tested. A simple bend is programmed on the fiber and the ability of the thin elongated object to transition from an arbitrary shape to the programmed shape is studied. In particular, we monitor the recovered angle $R_n$ as a function of the cycle number $n$ (Fig. 5a), i.e. the percentage of the initially programmed angle which is recovered after a given number of bending/straightening cycles. More details for $R_n$ calculation are provided in the methods section. In practice, the initial angle $\alpha_{prog}$ is measured on the sample just after the programming stage (described in Fig. 3). Then, the fiber is straightened at room temperature, i.e. at a temperature below the glass transition temperatures of the various polymers involved in the different fiber architectures. The sample is then immersed in hot water (≈80 °C) to retrieve the initially programmed bent shape, which completes a full cycle. The obtained angle is measured and the process is repeated $n$ times. To test its reproducibility, the experiment is first carried out on five samples with similar attributes (diameter of 450 μm) and with a programmed angle of 90° (see Fig. 5b). After the very first cycle, the different samples only restore 90 to 70% of the programmed curvature. After that, the value of the recovered angle keeps on decreasing slowly over the next cycles, with a dispersion of about ±7% (average standard deviation) around the mean value. After 25 cycles, the fibers have recovered, on average, 53 ± 7% of the initially programmed angle. Intervals corresponding to one, two, and three standard deviations of the mean are plotted on the graph, where respectively 68 %, 95% and 99.7% of the data points should fall. The impact of sample dimensions is also tested as described in Fig. 5c where shape recovery experiments carried out on fibers of various diameters are recapped. No particular correlation between the recovered angle and the fiber diameter is evidenced here. Samples with 450 μm diameter perform the best while samples of 220 μm and 900 μm diameter perform very similarly. Those results are in fact consistent with the recovery angle reproducibility range discussed just before. Additional tests are then performed on the SM-sleeve as its purpose is to be combined with commercial fibers.

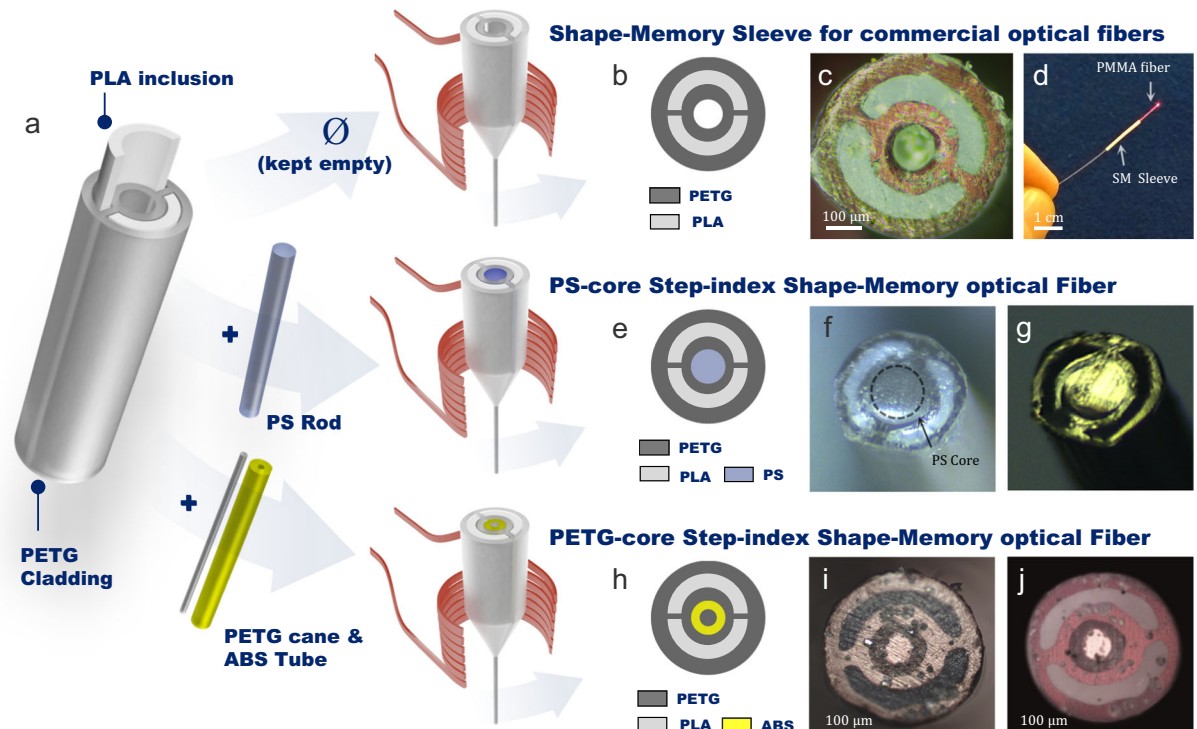

**Fig. 4 | Design and fabrication of shape-memory optical fibers. a** 3D diagram of the Polylactic acid (PLA)/polyethylene terephthalate glycol (PETG) frame which is considered for shape-memory fibers production. Three different architectures are produced from this main assembly: **b** 2D diagram and **c** optical microscope image of the cross-section of a fibered hollow sleeve which **d** can be later associated with a commercial optical fiber, herein a Polymethyl methacrylate (PMMA) fiber. **e** 2D diagram of the cross-sectional profile of the SM-PS fiber. Cross-sectional view of the fiber taken with a microscope in **f** reflection mode and **g** transmission mode. **h** 2D diagram of the cross-sectional profile of the SM-PETG fiber. Cross-sectional view of the fiber taken with a microscope in **i** reflection mode and **j** transmission mode.

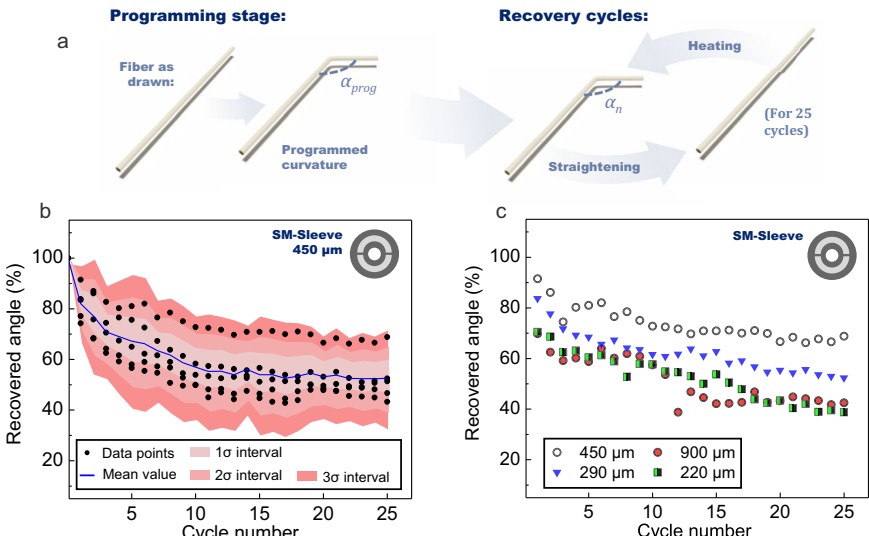

**Fig. 5 | Shape-recovery performances of a hollow shape-memory fiber.**
**a** Description of the shape-memory experiment: $\alpha_{prog}$ is the programmed angle and
$\alpha_n$ is the angle measured at cycle number $n$. **b** Recovered angle (in percentage) as
function of cycle number for five similar SM-sleeve samples and for a programmed

angle of ≈90°. The mean value of the recovered angle (blue line) and the standard
deviation (σ) intervals (red bands) are also plotted on the graph. **c** Recovered angle
(in percentage) as function of cycle number for SM-sleeve samples with different
external diameters and for a programmed angle of ≈90°.

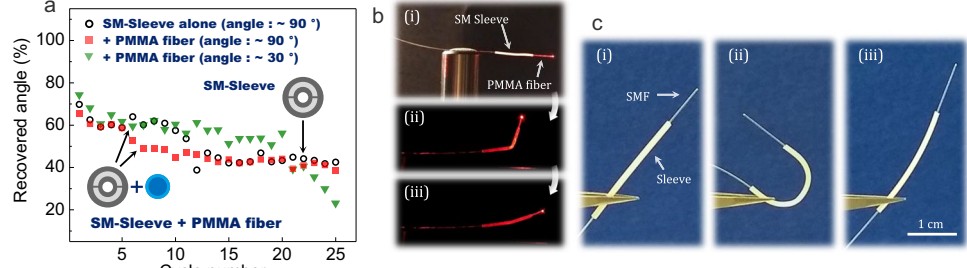

**Fig. 6 | Shape-recovery performances of a hollow shape-memory fiber when
combined with commercial fibers. a** Recovered angle (in percentage) as a func-
tion of cycle number for a ≈ 900 μm external diameter SM-sleeve with or without a
Polymethyl methacrylate (PMMA) fiber inserted in its central hole and for a pro-
grammed angle of either ≈30° or ≈90°. **b** Top to bottom: picture of an SM-sleeve

sample associated with a PMMA fiber, picture of the assembly in a temporary shape,
and when it has recovered its original straight shape. **c** (i) a picture of an SM-sleeve
assembled with a silica SMF, (ii) a picture of the assembly in a temporary, and (iii) a
recovered configuration.

Specialty active coatings or sheaths appear as a serious alternative
approach to the direct thermal drawing for the implementation of
additional functionalities to elongated objects such as fibers[6,7,36] or
even thin metallic wires[8]. In general, hollow-fiber structures can be
used to create microchannels for microfluidics or guides for endo-
scopic tools for remote interventions. Here, the SM-sleeve structure is
used to add a shape-memory function to a standard fiber. The size of
the central hole of the hollow structure is indeed tailored so that
commercial glass or polymer step-index waveguides can fit in it. Such
assemblies then benefit from the mature technology of standard
commercial fibers, especially in terms of optical properties (e.g., better
connectivity, low attenuation for silica and PMMA fibers or mid-
infrared transmission of fluoride or other fibers) as well as the high
versatility of elongated multimaterial structures.

The shape recovery performances of a segment of SM-sleeve
combined with a multimode PMMA fiber are tested over 25 cycles
(Fig. 6a). The internal diameter of the sleeve is ≈300 μm (corre-
sponding to ≈900 μm external diameter) so that the 250-μm-diameter
PMMA fiber can fit in the central channel. Fig. 6b shows pictures of
PMMA fiber combined with a 1-cm-long portion of SM-sleeve for dif-
ferent stages of the recovery process. The bending/straightening cycle
experiments are carried out for two programmed angles (30° and 90°)

and values of the recovered angle are compared to what is obtained for
a 900-μm-diameter SM-sleeve alone (i.e. without PMMA fiber). The
three distinct configurations follow a very similar trend, which reveals
that the combination with the PMMA fiber does not significantly
impact the shape-memory performances of the SM-sleeve. Fig. 6c
shows pictures of an SM-sleeve combined with a silica single-mode
fiber (SMF). In a similar fashion, the assembly of the SM-sleeve with
commercial silica SMF is able to transition from an arbitrary shape,
here a U-shape curvature (obtained via hot-forming) with a bending
radius of nearly 4 mm, back to its original straight shape. Though the
manipulation of glass fiber structures requires additional precautions
(limited bending radius, hot-forming preferred), those results confirm
that the use of hollow specialty sheaths can be applied to bring shape-
memory functions to standard commercial fibers.

The shape recovery experiments are now carried out on the all-
solid SM-PS and SM-PETG architectures and their performances are
compared to those of the SM-sleeve. For a 30° angle, the three fibers
behave very similarly (see Fig. 7a). The SM-PETG sample slightly out-
performs the two other structures as it retains more than 70% of the
initial shape after the 25 cycles against 63% and 57% for the SM-sleeve
and SM-PS respectively. For a 90° programmed angle (see Fig. 7b), as
already discussed, the SM-sleeve maintains good shape-memory

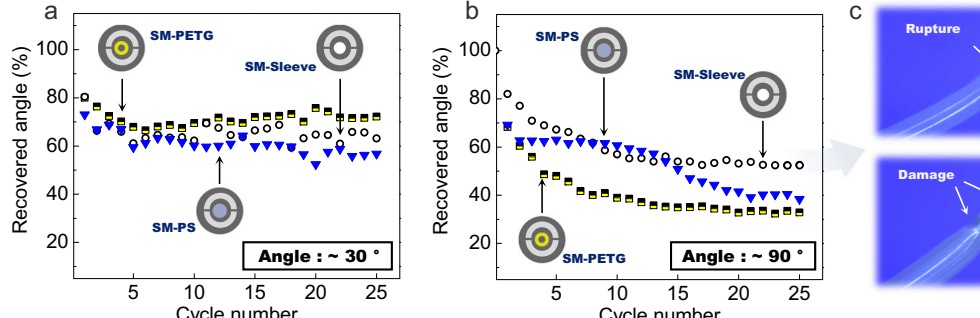

**Fig. 7 | Shape-recovery performances of shape-memory optical fibers.** Recovered angle (in percentage) as function of cycle number for an SM-sleeve (≈450 μm), an SM-PETG fiber (≈440 μm), and an SM-PS fiber (≈440 μm) and for a programmed angle of **a** ≈ 30° and **b** ≈ 90°. **c** Longitudinal view of SM-PS (top picture) and SM-PETG (bottom picture) samples after 25 bending/straightening cycles.

abilities with a ≈ 53% recovered angle at the end of the experiment. This is however not the case for the SM-PETG structure which now only restores 30% of the initial curvature after the 25 cycles (but remains constant after 15 cycles). In a different way, the SM-PS fiber exhibits a singular behavior that can be decomposed in two phases: first the structure performs similarly to other ones with a loss of ≈30% of the initial bend which then stabilizes at ≈60% over the first 10 cycles. After that, the decrease of the recovered angle accelerates around cycles 12-13 and the $R_n$ value ends up at 40% upon completion of the experiment. As expected, the nature of the thermoplastics involved in the elongated structure ultimately affects the mechanical behavior of the fiber. Here the sample which contains polystyrene is not able to sustain the entire shape recovery test, especially when considering deformations with important amplitude. Fig. 7c shows pictures of the damage induced to the SM-PS and SM-PETG fibers after 25 bending/straightening cycles and for a 90° programmed angle. The SM-PS sample is ruptured and only holds in place from a small portion of its section which remains intact. It is worth noting that the structure is still able to recover a non-negligible portion of the programmed angle despite being greatly damaged. In comparison, the SM-PETG sample suffers less deterioration and only exhibits signs of necking/tapering and local changes in density in the vicinity of the mechanically solicited zone. It is worth mentioning that the experiments are carried out here in extreme conditions for the thin elongated objects. The bending is performed at room temperature, which is below the glass transition temperatures of the various polymers involved in the fiber architecture and will induce severe alteration of their structure. Additionally, the bending radius is particularly small which implies large axial compressions and elongations. This induces important mechanical stress which intensifies when moving away from the longitudinal axis of the fiber. However, the ability of the fibers to still recover their programmed shape under such challenging conditions provides a significant advantage for the development of user-adjustable devices that need to be rectified on the go at room temperature.

As already mentioned, the viscoelastic properties of the considered polymers will greatly affect the behavior of the elongated structure. Polystyrene is a particularly brittle material[37], which means it does not deform in a plastic fashion or elongate before rupturing but rather forms cracks upon repeated mechanical stress and then abruptly breaks. It is therefore consistent to observe brittle-like fractures occurring for PS-containing samples when they are subjected to bending/straightening cycles. PLA and ABS are also classified as brittle, but they possess higher elongation at break and can sustain larger strains than PS. PETG on its part is a ductile material (with elongation at break as high as 240%) and can therefore better withstand the successive flexural tests[38]. Young's moduli as well as the elongation at the break of the thermoplastics selected here are recapped in Table 1. Taking into account those mechanical considerations, the choice of

the thermoplastics involved in the fiber architecture appears unsurprisingly crucial. Elastomers could be considered here to improve the mechanical sustainability of the fibers and thus their shape-memory performances. It has indeed been recently demonstrated that those polymers can be powerful materials for the development of highly integrated multimaterial fiber actuators with optical and electrical sensing capabilities[5].

A concerning issue is that the damage induced from the mechanical solicitations discussed just above may seriously hamper the light-transmission properties of the fabricated waveguides. For this purpose, we address now the impact of shape-memory cycling on the optical performances of our step-index fibers, as described in Fig. 8. The transmission of the SM-PETG and SM-PS structures in the visible range at 635 nm is monitored throughout a series of straightening/bending cycles. In practice, straight-as-drawn samples are selected and bent at room temperature to form a 90° angle. After that, they are heated up using a heating block to recover a rectilinear shape. At the same time, a measurement of the output power is performed when the fibers are in a straight or curved configuration. This operation is repeated up to 20 times and corresponding experimental results are recapped in Fig. 8c, d for the SM-PETG and SM-PS fibers, respectively. It is worth mentioning that loss measurements have been performed beforehand on each of the two fibered structures at 635 nm through the cutback technique (Fig. 8a, b). The all-solid step-index structures exhibit attenuations of ≈0.1 dB cm⁻¹ and 0.8 dB cm⁻¹ (for the SM-PS and SM-PETG respectively) which are typical losses for entry-level POF[39]. Figure 8e shows pictures of a light-guiding fiber completing a recovery cycle, i.e. transitioning from a temporary bent shape to a programmed rectilinear shape. The SM-PS fiber, when straightened, is able to maintain good transmission (≈90%) only on the first three cycles. In the bent configuration, maximum transmission only reaches ≈50% due to strong bending losses induced by the curvature. The bending radius is indeed in the order of magnitude of the fiber diameter, i.e. millimetric. Then, transmission in the rectilinear shape monotonously decreases, and around cycle 13, the fiber completely ruptures and light is not propagated anymore through the thin waveguide. Before that, around cycle 9, the transmitted power through the fiber curved at a 90° angle had already abruptly fallen below 10%. Cycle 9 seems in fact to mark the starting point of irreversible defects or cracks formation within the polystyrene core. The SM-PETG sample on its part performs significantly better and is able to maintain, in the rectilinear configuration, 90% of transmission after 18 cycles. After that, the transmitted power decreases drastically due to fatigue-related damage induced to the light-guiding core. In the bent configuration, the SM-PETG sample preserves a good transmission of 50–70% also up to the 18th test iteration. Overall, because of the higher core-cladding refractive index difference, the SM-PETG architecture is less sensitive toward bending losses. Those experiments demonstrate that our fibers simultaneously

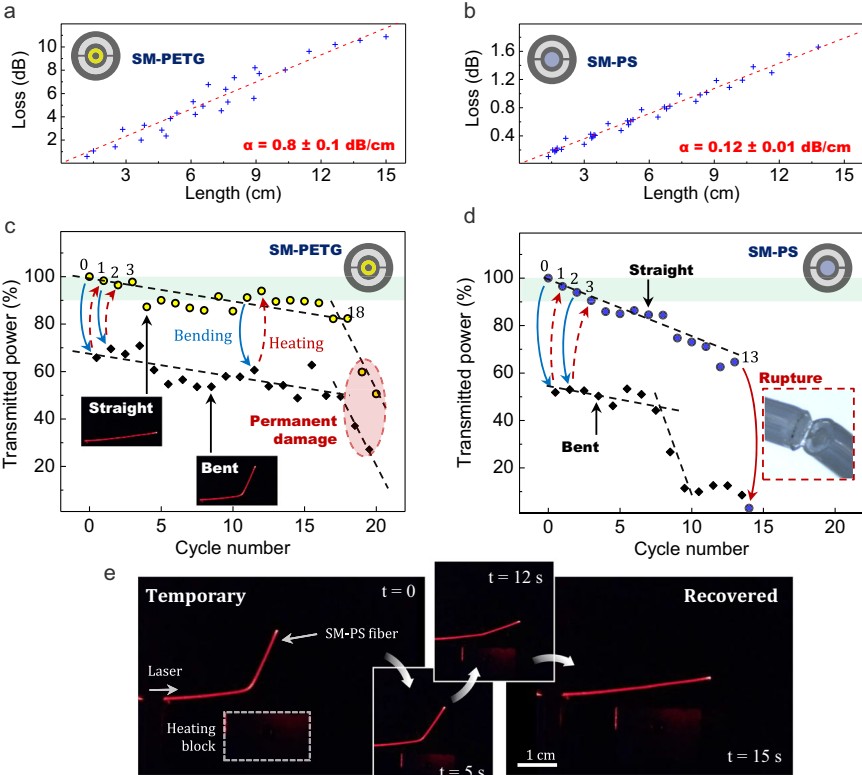

**Fig. 8 | Optical properties of shape-memory optical fibers before and during shape-recovery cycles. a**, **b** optical attenuation measurements at 635 nm of the SM-PETG and SM-PS fibers respectively. **c** Transmitted power at 635 nm as a function of cycle number for a 450 μm outer diameter SM-PETG fiber. Yellow circles refer to measurements performed when the fiber has a straight shape (see left inset) and black squares when it is bent (see right inset). **d** Transmitted power at 635 nm

as a function of cycle number for a 440 μm diameter SM-PS fiber. Blue circles refer to measurements performed when the fiber has a straight shape and black squares when it is bent. Inset: Picture of the ruptured SM-PS fiber after 15 cycles. **e** Performance of a shape-memory optical fiber (SM-PS). From left to right: the fiber is in a temporary shape and progressively recovers its original straight shape when heated.

hold shape-memory abilities and light guiding function, which has great potential for the development of probes or soft actuators with light delivery or optical sensing capabilities and others. Due to the good resistance of polymers towards acids and bases, the device can operate in a wide range of environments including biological media, rendering the present fiber attracting for applications in endoscopy. An SM-PETG fiber is then tested to transport a 635 nm laser beam toward a target object placed behind an obstacle (herein a black screen). The shape-memory fiber is heated to recover its programmed shape and is able to deliver the optical signal at a ≈170° angle from its original direction. The experimental operation is depicted in Fig. 9a. To minimize the bending losses, a curve with a 2.5-mm-bending radius is programmed to the sample. We confirm that shape-memory fibers can be successfully employed here to circumvent obstacles and illuminate hard-to-reach targets, which opens the way to new light delivery strategies in challenging and exiguous environments.

As a final experiment, a shape-memory exposed-core fiber is fabricated to perform fiber evanescent wave spectroscopy (FEWS) proof-of-principle measurements. The hemicylindrical geometry of the fiber is derived from the profile described in Fig. 4h and features a PETG core surrounded by an ABS half-cladding (see inset on Fig. 9b). A 15 cm sample of this fiber is programmed into a U-shape with an 11 mm bending radius following the procedure detailed in Fig. 3a. White light from a LED is coupled at one end of the sample and the other extremity is cold-spliced to a 100 μm core silica multimode fiber (MMF) for signal collection purposes. The MMF then sends the optical information towards a fibered spectrometer operating in the 400–900 nm range. Transmission in the visible range of the shape-memory exposed-core fiber is then monitored when transitioning from a straight shape to a

180° U-shape and exposed to a dye (herein a water-based blue ink). Absorption spectra of the coated sample, when oriented at different angles, are given in Fig. 9b. The dynamic response of the sensor is also tested by measuring the real-time evolution of the absorption at 635 nm when a fiber sample transitions from a straight shape to a 180° U-shape (see Fig. 9c). The data are recorded using a red Laser source as the incident light and a silicon photodiode as the detection apparatus. Response of the device is quick as a full recovery cycle is performed in approximately 40 s and can be paused at a chosen angle as depicted by the absorption steps in Fig. 9c. On Fig. 9b, the background attenuation at 450 nm monotonously increases as function of the angle due to stronger guiding losses. The amplitude of the two main absorption bands of the dye however increases before reaching a saturation. This saturation can be detrimental to precise measurements of analyte concentrations using FEWS detection scheme. Amplitude of the absorption band centered at 645 nm (i.e. the different between the background attenuation at 450 nm and the maximum attenuation at 645 nm) as function of the orientation angle is plotted in Fig. 9d. It clearly shows a maximum around 90° which therefore stands out as the optimal angle to perform FEWS measurements in the present configuration, as maximization of the absorption amplitude is particularly useful to improve the sensitivity of the sensor. Those proof-of-principle experiments illustrate the ability of the proposed device to dynamically adapt and optimize the shape of the sensing fiber. This can prove particularly useful to improve detection using FEWS techniques as the precision of the measurement depends on a wide variety of parameters (main absorption wavelengths range, evanescent-wave penetration depth, angle of attenuated total reflectance, concentration and refractive index of the analyte, etc.).

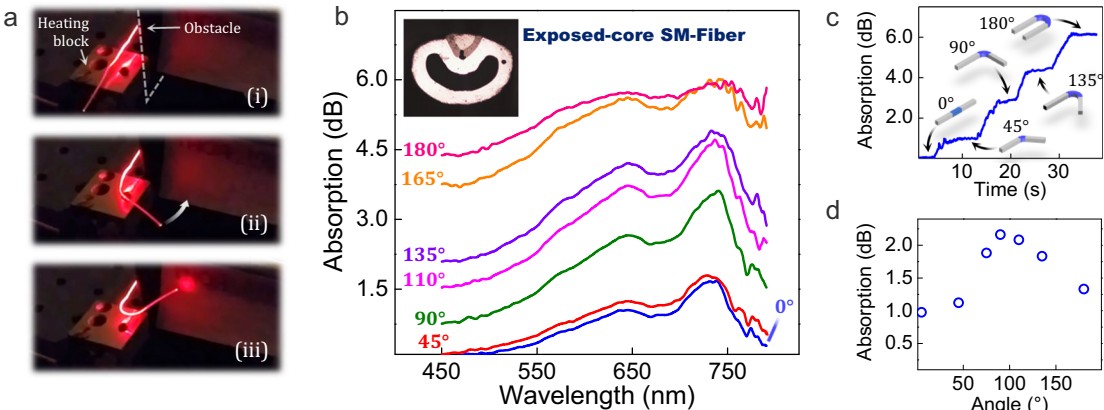

**Fig. 9 | Proof of principle experiments of an adaptative fiber evanescent wave spectroscopy (FEWS) sensor using a shape-memory fiber. a** SM-PETG fiber delivers light around an obstacle and is shown (i) in a temporary straight configuration, (ii) in transition, and (iii) in its programmed shape forming a ≈170° U-shape. **b** Visible absorption spectra of a blue ink-coated exposed-core shape-memory fiber when bent with different angles. Inset: a cross-sectional view of the fiber in transmission mode. **c** Evolution in time of the absorption of the coated fiber at 635 nm when going from a straight shape to a programmed U-shape. **d** Amplitude of the 645 nm absorption band as function of orientation angle.

The integration of fiber technology and 4D-printing could enable new possibilities for the application of multimaterial fibers in particular, which already find extensive use in photonics, electronics, and robotics[3]. Microfibers are for instance woven within fabrics to embed advanced sensing or energy-gathering devices into clothes[2]. The present findings could specifically contribute to the development of shape-changing smart textiles for lighting/display devices and wearable garments capable of monitoring health[40,41]. These research advancements also extend to the field of soft robotics, introducing novel photo-actuators that can be remotely controlled using light. An exemplary outcome of our findings could for instance be the creation of micro-grippers or more generally micro-actuators that exhibit responsive behavior to light stimuli. Shape-memory fibers are expected to find application in the health industry where device miniaturization is sought. The proposed methodology is highly adapted to the development of evolutive implantable biological photonic devices and could help address specific requirements in this field such as wave-guiding properties (e.g. production of the slab, ridged, channel waveguides, etc.) and adaptable optical features (e.g., tuning of internal scatterers, switching from side emission to end emission, etc.)[24]. The fact that the proposed fibers simultaneously hold shape-memory abilities and light-guiding function also means that they hold great potential for the development of endoscopic probes for vascular or catheter procedures where navigation in complex exiguous environments is required. Finally, as demonstrated through the proof-of-principle FEWS measurement, our SM-waveguides could contribute to the progress of adaptable fiber-based micro-optics and optical sensors based on techniques such as absorption spectroscopy and detection of refractive index change.

In the present work, we demonstrated, for the first time, that tens-of-meter of thin microstructured light-guiding objects with shape-memory capabilities can be produced from the thermal stretching of a single polymeric preform. We show that a simple hollow architecture can be combined with commercial fibers to add to them a shape-memory function. Two all-solid polymer fiber profiles are also produced using the proposed methodology and their optical as well as actuation properties are assessed. Light transmission is demonstrated in fiber samples which are able to transition from rectilinear shapes to curved shapes forming angles up to 170 ° and with a bending radius of ≈2.5 mm. In addition, an exposed-core fiber architecture with shape memory abilities is produced and successfully employed in proof-of-principle FEWS experiments further proving the potential or our strategy for the development of adaptable in-fiber functionalities. Future works will be devoted to the optimization of the fiber geometry to improve both the optical and mechanical behavior of the proposed shape-memory objects. In particular, the PETG/PLA ratio will be optimized and honeycomb structures with better flexibility will be explored. In conclusion, the integration of shape-memory materials within fibers through the preform-to-fiber drawing process is expected to enable the scalable production of highly compact and dexterous devices with optical delivery and sensing abilities.

## Methods

### Preform preparation and fiber drawing

Preforms were assembled from additively manufactured parts. Commercial (Ultimaker) 2.85 mm PLA (white) and PETG (transparent) filaments were used here. Note that the filaments are stored in a vacuum when not in use to avoid moisture contamination of the thermoplastics. The different pieces are printed separately using an entry-level FDM desktop printer (Ultimaker S3) with a 0.4 mm diameter standard brass nozzle. The main printing parameters are as follows: 0.2 mm layer height, 0.4 mm line width, and 60 mm s$^{-1}$ printing speed. The fabricated parts are then dried at 60 °C in a vacuum oven for 24 h hours after what they are put together to form the preform. The assembly is then thermally stretched using a dedicated 3m-high draw tower: the preform is placed inside the furnace of the draw tower of which the temperature is slowly ramped up (10 °C min$^{-1}$). To avoid any contamination, a 3 L min$^{-1}$ He gas flow is applied during the procedure. After the initiation of the elongation process, the preform is slowly fed into the furnace while the drawing parameters are continuously monitored to produce the fiber with the desired dimensions.

### Differential Scanning Calorimetry (DSC)

DSC experiments were performed on a TA Instruments Q1000 system. 5 to 10 mg samples were placed in standard aluminum pans and heated at 10 °C min$^{-1}$ up to 200 °C under nitrogen gas flow and a last 5 min isothermal step. The glass transition temperature $T_g$, the melting temperature $T_m$, and the crystallization temperature $T_c$, were determined using Thermal Analysis™ software.

### Recovery angle measurements

The shape-memory capabilities of the fibers were determined by measuring the recovery rate $R_n$ in % of a programmed angle for a given cycle number $n$. In practice, the programmed angle $\alpha_{prog}$ and the

recovered angle $\alpha_n$ were measured through digital analysis (using ImageJ software) of pictures of the fibers at the different stages of the bending/straightening process. $R_n$ was then calculated using the following equation:

$$R_n = \frac{\alpha_n}{\alpha_{prog}} \times 100 \qquad (1)$$

**Transmission properties of the shape-memory fiber**
Optical loss measurements were carried out using the cutback method on a few tens of centimeter-long fiber samples. A 635 nm continuous-wave laser source with 6-mW average power and single-mode silica fiber output is butt-coupled to the polymer fiber. The transmitted power of the shape-memory fiber is then measured for different sample lengths using a Si photodiode power sensor.

Power transmission as a function of the shape-recovery cycle is determined as follows: the 635 nm continuous-wave laser source is butt-coupled to the shape-memory polymer fiber. The power transmitted through the SM fiber is measured using a Si photodiode power sensor when it is in its programmed rectilinear shape. The fiber is then bent at room temperature to form a near 90° angle and the transmitted power is measured again. Following, the fiber is heated to recover its original straight shape, which completes a single cycle. The transmitted power is measured again and the deformation cycle of the fiber is repeated.

## Data availability
The data that support the findings of this study are available from the corresponding author upon request.

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

## Acknowledgements
This research was funded by from French program Investments for the Future operated by the National Research Agency (ISITE-BFC, 4DMeta project, contract ANR-15-IDEX-03; and EIPHI Graduate School, SMILE project, contract ANR-17-EURE-0002). The authors also acknowledge support from the French ANR (through the TRAFIC project, contract ANR-18-CE08-0016-03), from CNRS Institute of Physics (Emergence 2017, 3D-printed optical fibers), and from European Regional Development Fund. This work benefited from the facilities of the SMARTLIGHT platform funded by the French ANR (EQUIPEX+ contract ANR-21-ESRE-0040) and the Région Bourgogne Franche-Comté.

## Author contributions
C.S., M.E., F.D., G.G., J.-C.J., and C.-H. B. conducted the investigations and performed data collection, curation, and analysis. C.S., F.D., C.-H.B., B.K., and F. S. wrote the original manuscript.

## Competing interests
The authors declare no conflict of interest.
