## [Peer Review File · Nature Communications]

4D Optical Fibers based on shape-memory polymersREVIEWER COMMENTS

Reviewer #1 (Remarks to the Author):

This paper about shape-memory fibers is interesting and well written, and could probably lead to interesting applications. I would suggest a round of revisions to improve its content,

1. The resolution of some figures could be improved. Also, all the info presented is not always pertinent, with too many details or subfigures without any relevant input.
2. An interesting study would be to characterize the behavior of these structures in other media or solutions, to show or tune its resistance to certain chemicals or pollutants.
3. The discussion about the recovery after multiple cycles is interesting but leads to poor recovery results. The authors should discuss the possibility to improve the methodology to reach higher reproducibility over multiple cycles.
4. Some graphs do not present legend for the symbols on the curves. Also, there is a high disparity from one sample to another. more statistical analyses should be provided to study the significancy.
5. More final applications for such materials could be thoroughly discussed in the paper to emphasize the relevance of this research.

Reviewer #2 (Remarks to the Author):

This manuscript reported the manufacturing of shape-memory optical fibers from the thermal stretching of additively manufactured preforms and the detailed study on the shape-memory abilities and light guiding under various configurations. The reported results are interesting to the field, and the manuscript is clearly written. However, my main concern is the novelty of this work, as well as the doubt about the optical characterization. Based on the following comments, I do not believe this work can meet the criteria of Nature Communications.

1. One critical flaw is that the work is not placed in the context of the extensive existing literature on the fabrication of optical fibers from additively manufactured preforms, which has been well studied for quite a while. I cannot see any fundamental difference between the previously published works and this submission, even though this submission demonstrates the shape-memory function. The achieved results are interesting, but not a major advance.
2. One of the major challenges of the additively manufactured preforms is the surface roughness, especially on the interface between core and cladding. This is also the reason for the high optical loss. From the cross-section images of the achieved optical fibers in this work, the waveguide quality is poor, such as deformed cores and rough interfaces between different materials. The loss should be high, for example, a lot of light is scattered out, as shown in these light-guiding figures in the manuscript. However, the measured loss is only about 0.1 dB/cm, which is questionable.
3. The authors mentioned that they "strongly expect that such actuatable fibers with light-guiding abilities will trigger exciting progress of unprecedented smart devices in the areas of photonics, electronics, or robotics." But I really could not see how such optical fibers could trigger the exciting progress of unprecedented smart devices in the areas of photonics, electronics, or robotics.

Reviewer #3 (Remarks to the Author):

The manuscript titled "4D Optical Fibers based on shape-memory polymers" is correctly set up. This work is relatively interesting, but it does not see much practical application value. Besides, several problems should be concerned before publication. I would like to make a decision after the article goes through a major revision.

1. The author only briefly mentioned the crystalline properties and melting point of PLA, which are not sufficient to illustrate its shape memory properties. I did not find the principle of shape memory for PLA in the manuscript, and the author needs to supplement relevant information in detail.
2. The fiber diameter in Figure 4a is quite thick (about 500 μm). Is this the smallest diameter? How to further reduce the diameter of the fiber. Moreover, most experiments only test one fiber, but I'm afraid one fiber is not enough in practical applications. If many fibers are woven together or woven into a film, how will its shape memory be?
3. The most important issue is the lack of practical application of this fiber. A good material needs to be evaluated by actual application effects. It is recommended that the author supplement corresponding application cases.
4. The writing of the paper is too casual, for example, the first paragraph of 2.1 has only two sentences, and there are many popular science languages elsewhere. The overall logic of the paper is poor. Moreover, the uniformity of the graphics needs to be improved.

Reviewer #1 (Remarks to the Author):

This paper about shape-memory fibers is interesting and well written, and could probably lead to interesting applications. I would suggest a round of revisions to improve its content,

1. The resolution of some figures could be improved. Also, all the info presented is not always pertinent, with too many details or subfigures without any relevant input.

We thank the reviewer for their remark. For better clarity, we have reworked and simplified several of the Figures:

- Figure 4 layout was reworked and subfigure 4k was removed for simplification
- Unnecessary details were removed from Figure 5 b and c
- Missing legends were added in several Figures
- Subfigures 6 d and e were removed for simplification
- Subfigure 8e was simplified and Subfigure 8f was transferred to Figure 9

2. An interesting study would be to characterize the behavior of these structures in other media or solutions, to show or tune its resistance to certain chemicals or pollutants.

As mentioned by the reviewer, it would be interesting to assess the performances of our shape-memory fibers when exposed to specific environments. We expect the fiber resistance to be dependent on the cladding material properties, which is PETG in the case. The fibers will therefore be sensitive to organic solvents such as acetone, Tetrahydrofuran (THF) and so on. By contrast, the fibers should have good resistance towards acidic and basic media. Our fibers are therefore well adapted to endoscopic applications as they are highly likely to withstand exposure to body fluids or biological media.

We added the following sentence in the text (page 20):

“Due to the good resistance of polymers towards acids and bases, the device can operate in a wide range of environments including biological media, rendering the present fiber attracting for applications in endoscopy.”

3. The discussion about the recovery after multiple cycles is interesting but leads to poor recovery results. The authors should discuss the possibility to improve the methodology to reach higher reproducibility over multiple cycles.

In the manuscript, it was mentioned that the shape-recovery experiments were conducted under harsh conditions, which accounts for the observed reproducibility. The fibers were bent at room temperature, significantly below the glass transition temperatures of the various polymers present in the fiber structure. This bending caused structural changes in the polymers. Typically, shape memory objects are deformed around the glass transition temperature of the polymers involved [a]. The ability of our fibers to still recover their programmed shape under such challenging conditions provides a significant advantage for the development of user-adjustable devices. This advantage is particularly valuable in medical applications where practitioners need to adjust implants on the go and at room temperature. To enhance reproducibility, improvements can be made in the fiber profile in which the PETG/PLA ratio can be tailored for better recovery ratios. Additionally, honeycomb structuration of the cladding could be used for better flexibility of the fiber. Finally, other polymers could be involved in the fiber structure. In particular, elastomers hold great potential for the present topic as they exhibit adaptable actuation performances while still being adapted to the thermal drawing process [b]. Future works will be devoted to further optimization of the shape-memory function of the fiber.

[a] Bautista-Salinas, D., Abdelaziz, M. E., Temelkuran, B., Yeatman, E. M., Huins, C. T., & Rodriguez y Baena, F. (2022). Towards a Functional Atraumatic Self-Shaping Cochlear Implant. *Macromolecular Materials and Engineering*, 307(1), 2100620.

[b] Leber, A., Dong, C., Laperrousaz, S., Banerjee, H., Abdelaziz, M. E., Bartolomei, N., ... & Sorin, F. (2023). Highly Integrated Multi-Material Fibers for Soft Robotics. *Advanced Science*, 10(2), 2204016.

We added the following text in the manuscript:

- Page 17:

“However, the ability of the fibers to still recover their programmed shape under such challenging conditions provides a significant advantage for the development of user-adjustable devices that need to be rectified on the go at room temperature.”

- In the conclusion (page 24):

“In particular, the PETG/PLA ratio will be optimized and honeycomb structures with better flexibility will be explored.”

4. Some graphs do not present legend for the symbols on the curves. Also, there is a high disparity from one sample to another. more statistical analyses should be provided to study the significancy.

An initial statistical analysis is provided in Figure 5b. Here, we plot the percentage of the initially programmed angle which is recovered as function of the cycle number n , with n up to 25. The experiment is carried out on five samples with similar attributes (diameter, fiber profile), which produce the set of points ($5 \times 25 = 125$) plotted in Figure 5b.

The mean value of recovered angle and standard deviation is then calculated for each cycle, i.e. for 5 measurements. The mean value is materialized on the graph as the continuous blue line. We modified Figure 5 to feature more statistically accurate intervals, i.e. the interval corresponding to one, two and three standard deviations of the mean. According to the normal distribution theory, 68 %, 95 % and 99.7 % of the data points should fall within these first, second and third interval respectively. Note that those intervals are calculated for a given cycle number, i.e. for five points. The intervals give, at glance, a good prediction of the expected recovered angle as function of cycle number. We also added a legend in Figure 5 to describe the discussed intervals and plots.

Additionally, we have reworked and simplified several of the Figures for better clarity. Changes are recapped in **response 1**.

We added the following sentence in the text (page 13):

“Intervals corresponding to one, two and three standard deviations of the mean are plotted on the graph, where respectively 68 %, 95 % and 99.7 % of the data points should fall.”

5. More final applications for such materials could be thoroughly discussed in the paper to emphasize the relevance of this research.

The present work focusses on the integration of fiber technology and 4D-printing, enabling new possibilities for the application of multimaterial fibers in particular. These fibers already find extensive use in photonics, electronics, and robotics [a]. In the context of woven fibers, our work could specifically contribute to the development of shape-changing smart textiles for lighting/display devices and wearable garments capable of monitoring health. Our research advancements also extend to the field of soft robotics, introducing novel photo-actuators that can be remotely controlled using light. An exemplary outcome of our findings could for instance be the creation of micro-grippers that exhibit responsive behavior to light stimuli. Additionally, our work could contribute to the progress of adaptable fiber-based micro-optics and optical sensors based on techniques such as absorption spectroscopy and detection of refractive index change. The fact that the proposed fibers simultaneously hold shape-memory abilities and light guiding function also means that they hold great potential for the development of endoscopic probes for vascular or catheter procedures. Finally, the methodology is also highly adapted to the development of evolutive implantable biological photonic devices, and could help address specific requirements in this field such as waveguiding properties (e.g. production of slab, ridged, channel waveguides, etc.) and adaptable optical features (e.g., tuning of internal scatterers, switching from side emission to end emission, etc.) [b].

[a] Shen, Y., Wang, Z., Wang, Z., Wang, J., Yang, X., Zheng, X., ... & Zhang, T. (2022). Thermally drawn multifunctional fibers: Toward the next generation of information technology. *InfoMat*, 4(7), e12318.

[b] Pearson, S., Feng, J., & del Campo, A. (2021). Lighting the path: light delivery strategies to activate photoresponsive biomaterials in vivo. *Advanced Functional Materials*, 31(50), 2105989.

We added text in the manuscript to better discuss the final applications of our fibers (page 22-23):

“The integration of fiber technology and 4D-printing could enable new possibilities for the application of multimaterial fibers in particular, which already find extensive use in photonics, electronics, and robotics.³ Microfibers are for instance woven within fabrics to embed advanced sensing or energy gathering devices into clothes. The present findings could specifically contribute to the development of shape-changing smart textiles for lighting/display devices and wearable garments capable of monitoring health⁴¹. These research advancements also extend to the field of soft robotics, introducing novel photo-actuators that can be remotely controlled using light. An exemplary outcome of our findings could for instance be the creation of micro-grippers or more generally micro-actuators that exhibit responsive behavior to light stimuli. Shape-memory fibers are expected to find application in the health industry where device miniaturization is sought. The proposed methodology is highly adapted to the development of evolvable implantable biological photonic devices, and could help address specific requirements in this field such as waveguiding properties (e.g. production of slab, ridged, channel waveguides, etc.) and adaptable optical features (e.g., tuning of internal scatterers, switching from side emission to end emission, etc.).²⁴ The fact that the proposed fibers simultaneously hold shape-memory abilities and light guiding function also means that they hold great potential for the development of endoscopic probes for vascular or catheter procedures where navigation in complex exiguous environments is required. Finally, as demonstrated through the proof-of-principle FEWS measurement, our SM-waveguides could contribute to the progress of adaptable fiber-based micro-optics and optical sensors based on techniques such as absorption spectroscopy and detection of refractive index change.”

41. Choi, H. W. *et al.* Smart textile lighting/display system with multifunctional fibre devices for large scale smart home and IoT applications. *Nat. Commun.* 13, 814 (2022).

Reviewer #2 (Remarks to the Author):

This manuscript reported the manufacturing of shape-memory optical fibers from the thermal stretching of additively manufactured preforms and the detailed study on the shape-memory abilities and light guiding under various configurations. The reported results are interesting to the field, and the manuscript is clearly written. However, my main concern is the novelty of this work, as well as the doubt about the optical characterization. Based on the following comments, I do not believe this work can meet the criteria of Nature Communications.

1. One critical flaw is that the work is not placed in the context of the extensive existing literature on the fabrication of optical fibers from additively manufactured preforms, which has been well studied for quite a while. I cannot see any fundamental difference between the previously published works and this submission, even though this submission demonstrates the shape-memory function. The achieved results are interesting, but not a major advance.

Previous studies reported in literature concerning the fabrication of fibers from additively manufactured preforms mainly focus on the optical function (optical losses, waveguiding properties, etc.), which is not the case here. We believe the present work is a major advance for the following reason:

- We produce here shape-memory objects with unprecedented lengths (> 50 m here but potentially > 500 m) while still exhibiting controlled structuration at the micrometric scale. Structuration of shape memory objects through 4D-printing methods is usually limited to $10^3 - 10^5$ times the printing resolution, due to the size of the printing space of the used printer.
- The elaboration of those shape-memory objects is also done from a single step, highly scalable straightforward procedure (as opposed to the post-drawing strategies described in literature [a, b]) which is the preform-to fiber drawing process.
- The fabricated fibers moreover possess light-guiding abilities, hence their potential for applications in photonics.

[a] He, Y., Liang, H., Chen, M., Jiang, L., Zhang, Z., Heng, X., ... & Yang, Z. (2021). Optical Fiber Waveguiding Soft Photoactuators Exhibiting Giant Reversible Shape Change. *Advanced Optical Materials*, 9(21), 2101132.

[b] Cheng, J., Zhang, L., Zhao, K., Wang, Y., Cao, X., Zhang, S., & Niu, W. (2021). Flexible Multifunctional Photonic Crystal Fibers with Shape Memory Capability for Optical Waveguides and Electrical Sensors. *Industrial & Engineering Chemistry Research*, 60(23), 8442-8450.

We added a paragraph in the introduction to better contextualize our work regarding the manufacturing of preforms using 3D printing (page 4):

“Air/polymer structures were initially developed due to their simplicity but with the improvement of additive manufacturing technologies,¹⁶ more intricate architectures could be implemented including hollow-core profiles or non-cylindrical geometries.^{17,18} A lot of efforts have specifically been deployed for the fabrication of fibers with good optical quality using the technique, i.e. with defect-free structures and low attenuation levels.¹⁹ The range of materials used has also considerably widened, now encompassing doped polymer matrices or even inorganic compounds such as soft glasses and the all-important silica.^{13,20,21}”

16. Cook, K. *et al.* Air-structured optical fiber drawn from a 3D-printed preform. *Opt. Lett.* 40, 3966 (2015).

17. Talataisong, W. *et al.* Mid-IR Hollow-core microstructured fiber drawn from a 3D printed PETG preform. *Sci. Rep.* 8, 8113 (2018).

18. Zhao, Q. *et al.* Optical fibers with special shaped cores drawn from 3D printed preforms. *Optik (Stuttg)*. 133, 60–65 (2017).

19. Luo, Y., Canning, J., Zhang, J. & Peng, G.-D. Toward optical fibre fabrication using 3D printing technology. *Opt. Fiber Technol.* 58, 102299 (2020).

20. Chu, Y. *et al.* Silica optical fiber drawn from 3D printed preforms. *Opt. Lett.* **44**, 5358 (2019).

21. Kaufman, J., Bow, C., Tan, F. & Abouraddy, A. C. and A. 3D Printing Preforms for Fiber Drawing and Structured Functional Particle Production. in *Photonics and Fiber Technology 2016 (ACOFT, BGPP, NP) AW4C.1* (Optical Society of America, 2016).

2. One of the major challenges of the additively manufactured preforms is the surface roughness, especially on the interface between core and cladding. This is also the reason for the high optical loss. From the cross-section images of the achieved optical fibers in this work, the waveguide quality is poor, such as deformed cores and rough interfaces between different materials. The loss should be high, for example, a lot of light is scattered out, as shown in these light-guiding figures in the manuscript. However, the measured loss is only about 0.1 dB/cm, which is questionable.

The level of attenuation that we report here is indeed lower than state-of-the-art fibers produced from 3D-printed preforms which attain ~0.4 dB/cm [a]. The reason for this is that the polymer piece used in the preform is not 3D-printed (typically a filament used for 3D printing).

We clarified this subject in the manuscript and added the following sentence (page 11):

“Note that the central piece used for the core is not 3D-printed, which helps in minimizing optical attenuation.”

[a] Luo, Y., Canning, J., Zhang, J., & Peng, G. D. (2020). Toward optical fibre fabrication using 3D printing technology. *Optical Fiber Technology*, *58*, 102299. [c] Fu, R., Luo, W., Nazempour, R., Tan, D., Ding, H.,

3. The authors mentioned that they “strongly expect that such actuatable fibers with light-guiding abilities will trigger exciting progress of unprecedented smart devices in the areas of photonics, electronics, or robotics.” But I really could not see how such optical fibers could trigger the exciting progress of unprecedented smart devices in the areas of photonics, electronics, or robotics.

The present work focusses on the integration of fiber technology and 4D-printing, enabling new possibilities for the application of multimaterial fibers in particular. These fibers already find extensive use in photonics, electronics, and robotics [a]. We added text in the manuscript to better discuss the final applications of our fibers. See **Response 5 to reviewer 1** for more details.

We supplemented an application case of our fiber, which describes the development of an adaptative sensor based on evanescent wave absorption spectroscopy (FEWS). We produced an exposed-core fiber capable of transitioning from a straight shape to a programmed 180° U-shape. This helps to dynamically optimize the shape of the FEWS sensor which performances depend on a wide set of opto-geometrical parameters such as wavelength absorption range, refractive index and concentration of the analyte as well as the evanescent wave penetration depth or the angle of attenuated total internal reflectance. We demonstrate in a proof-of-principle experiment that the optimal detection angle is approximately 90° for the considered analyte (blue dye). This experiment shows the potential of our fibers for the development of evolutive compact sensors that can be optimized to a given situation (nature and concentration of analyte, wavelength detection range, etc.) thanks to the shape-memory abilities of the waveguides.

[a] Shen, Y., Wang, Z., Wang, Z., Wang, J., Yang, X., Zheng, X., ... & Zhang, T. (2022). Thermally drawn multifunctional fibers: Toward the next generation of information technology. *InfoMat*, *4*(7), e12318.

The corresponding description of the proposed sensor is given in the last paragraph of the manuscript (page 21-22) along with Figures describing its performances (See Figure 9b-d):

“As a final experiment, a shape-memory exposed-core fiber is fabricated to perform fiber evanescent wave spectroscopy (FEWS) proof-of-principle measurements. The hemicylindrical geometry of the fiber is derived from the profile described in Figure 4h and features a PETG core surrounded by an ABS half-cladding (see inset on Figure 9b). A 15 cm sample of this fiber is programmed into a U-shape with a 11 mm bending radius following the procedure detailed in Figure 3a. White light from a LED is coupled at one end of the sample and the other extremity is cold-spliced to a 100 μm core silica multimode fiber (MMF) for signal collection purposes.

Figure 9. (a) SM-PETG fiber delivers light around an obstacle and is shown (i) in a temporary straight configuration, (ii) in transition, and (iii) in its programmed shape forming a $\approx 170^\circ$ U-shape. (b) Visible absorption spectra of a blue ink-coated exposed-core shape-memory fiber when bent with different angles. Inset: cross-sectional view of the fiber in transmission mode. (c) Evolution in time of the absorption of the coated fiber at 635 nm when going from a straight shape to a programmed U-shape. (d) Amplitude of the 645 nm absorption band as function of orientation angle.

The MMF then sends the optical information towards a fibered spectrometer operating in the 400-900 nm range. Transmission in the visible range of the shape-memory exposed-core fiber is then monitored when transitioning from a straight shape to a 180° U-shape and exposed to a dye (herein a water-based blue ink). Absorption spectra of the coated sample when oriented at different angles are given in Figure 9b. The dynamic response of the sensor is also tested by measuring the real-time evolution of the absorption at 635 nm when a fiber sample transitions from a straight shape to an 180° U-shape (see Figure 9c). The data are recorded using a red Laser source as the incident light and a silicon photodiode as the detection apparatus. Response of the device is quick as a full recovery cycle is performed in approximately 40 seconds and can be paused at a chosen angle as depicted by the absorption steps on Figure 9c. On Figure 9b, the background attenuation at 450 nm monotonously increases as function of the angle due to stronger guiding losses. The amplitude of the two main absorption bands of the dye however increases before reaching a saturation. This saturation can be detrimental for precise measurements of analyte concentrations using FEWS detection scheme. Amplitude of the absorption band centered at 645 nm (i.e. the difference between the background attenuation at 450 nm and the maximum attenuation at 645 nm) as function of the orientation angle is plotted in Figure 9c. It clearly shows a maximum around 90° which therefore stands out as the optimal angle to perform FEWS measurements in the present configuration. Those proof-of-principle experiments illustrate the ability of the proposed device to dynamically adapt and optimize the shape of the sensing fiber. This can prove particularly useful to improve detection using FEWS techniques as the precision of the measurement depends on a wide variety of parameters (main absorption wavelengths range, evanescent-wave penetration depth, angle of attenuated total reflectance, concentration and refractive index of the analyte, etc.).”

Reviewer #3 (Remarks to the Author):

The manuscript titled "4D Optical Fibers based on shape-memory polymers" is correctly set up. This work is relatively interesting, but it does not see much practical application value. Besides, several problems should be concerned before publication. I would like to make a decision after the article goes through a major revision.

1. The author only briefly mentioned the crystalline properties and melting point of PLA, which are not sufficient to illustrate its shape memory properties. I did not find the principle of shape memory for PLA in the manuscript, and the author needs to supplement relevant information in detail.

We thank the reviewer for their question. The shape-memory mechanism in PLA is explained at the macromolecular scale by the presence in the polymer structure of netpoints and switchable segments. Netpoints are rigid domains which can be microscopic phases or crosslinking points that are responsible for the permanent shape fixation. Switchable segments are deformable domains as for instance amorphous phase. Above T_m , crystalline phases are melted and chain mobility is high so that the polymer can be easily shaped. When the material is cooled down and strained, i.e. held in a user-defined shape, netpoints and switchable segments form which fixates a programmed configuration. At temperatures close to T_g , netpoints stay in place but switchable segments possess enough chain mobility so that they can be altered (stretched, bent, etc.) and the polymer can be deformed into temporary configurations. Mechanical stress is however stored in those domains during that procedure. When the material is heated above T_g , the switchable domains gain mobility and release the stored energy by recoiling, i.e. by recovery the permanent shape.

We added text in the manuscript to better explain the observed shape-memory effects (page 7-9):

“The characteristic temperatures of PETG and PLA are extracted from the DSC curves plotted in Figure 1h and recapped in Table 1. Next, we evaluate the shape-memory performances of fabricated fibers. **By common acknowledgement, the practical strategies to control shape-memory effects in thermoplastics is to take advantage either of the glass transition temperature T_g or of the melting temperature T_m of the ordered phases³⁷. At the macromolecular scale, the shape-memory behavior is attributed to the presence of netpoints and switchable segments in the structure²⁸. Netpoints are rigid domains that can be microscopic phases, crosslinking points or entanglements and are responsible for permanent shape fixation³⁸. Switchable segments, on the other hand, are deformable domains, such as the amorphous phase. In PLA, above T_m , the crystalline phases melt, and chain mobility increases, allowing the polymer to be easily shaped. The programming stage consists in cooling down the material while applying a strain, i.e. holding the polymer in a user-defined shape. During this procedure, netpoints and switchable segments form which fixates the programmed configuration. At temperatures close to T_g , netpoints remain in place while switchable segments possess enough chain mobility to be altered (stretched, bent, etc.), enabling the polymer to be deformed into temporary configurations. However, mechanical stress is stored in these domains during the process. When the material is heated above T_g , the switchable domains gain mobility and release the stored energy by recoiling, thereby recovering the permanent shape.** PLA is heated above its melting temperature during the thermal elongation process and rapidly cools as it passes the neckdown region of the preform, thus solidifying. This procedure is equivalent to a programming stage for the shape-memory polymer. As the fiber is pulled vertically on the drawing tower, the shape which is programmed after the thermal drawing is a straight line. First shape recovery tests are performed on the as-drawn fibers, as described in **Figure 2**. Here, a ≈ 7 -cm-long portion of PETG/PLA fiber is deformed at room temperature to obtain a temporary shape which is in that instance a square spiral (see top left picture in Figure 2). **For the PLA and PETG matrices, the strain is thus fixed under their T_g , meaning that residual chain mobility in the amorphous domains authorizes deformation of the materials even at room temperature. When the manually applied stress is then removed at room temperature, the PETG/PLA fiber hold its temporary shape as long as the T_g of the PLA core and PETG cladding are not exceeded.** The shape-memory fiber is then immersed in deionized water at ≈ 80 °C (above the T_g values of PLA and PETG) without any applied stress, and is able to deploy into its original straight shape in a few tens of seconds (see bottom left picture in Figure

2). It is worth mentioning that no material flowing is observed due to the physical crosslinking of PLA by crystalline parts and the highly viscous amorphous state of PETG. “

37. Yakacki, C. M. *et al.* Strong, Tailored, Biocompatible Shape-Memory Polymer Networks. *Adv. Funct. Mater.* **18**, 2428–2435 (2008).

38. Xia, Y., He, Y., Zhang, F., Liu, Y. & Leng, J. A Review of Shape Memory Polymers and Composites: Mechanisms, Materials, and Applications. *Adv. Mater.* **33**, 2000713 (2021).

2. The fiber diameter in Figure 4a is quite thick (about 500 μm). Is this the smallest diameter? How to further reduce the diameter of the fiber. Moreover, most experiments only test one fiber, but I'm afraid one fiber is not enough in practical applications. If many fibers are woven together or woven into a film, how will its shape memory be?

Thermal drawing is a homothetic process where a macroscopic preform is stretched down into meters of thin fiber, which diameter can be arbitrary controlled in the range of $\sim 50 - 2000 \mu\text{m}$. In the present work, fibers with diameter of 220, 290, 450 and 900 μm were produced and tested for their shape-memory properties (see Figure 5c). Here, we specifically manufactured 450 μm hollow fibers possessing internal diameter of 150 μm that could host 125 μm diameter commercial fibers inside them.

We did not test the shape memory performances of multiple woven fibers which will be the purpose of future work. Still, we added text in the manuscript to better discuss this interesting application prospect for our fibers (see **response 5 of reviewer 1**).

3. The most important issue is the lack of practical application of this fiber. A good material needs to be evaluated by actual application effects. It is recommended that the author supplement corresponding application cases.

We thank the reviewer for their suggestion. We supplemented an application case of our fiber, which describes the development of an adaptative sensor based on evanescent wave absorption spectroscopy (See **answer 2 to reviewer 2**).

Additionally, we added text in the manuscript to better discuss the final applications of our fibers (see **response 5 of reviewer 1**).

4. The writing of the paper is too casual, for example, the first paragraph of 2.1 has only two sentences, and there are many popular science languages elsewhere. The overall logic of the paper is poor. Moreover, the uniformity of the graphics needs to be improved.

We thank the reviewer for their remark. The overall logic of the present paper can be summarized as follow:

- 1) The possibility to produce shape-memory fibers through the thermal drawing process is assessed
 - In particular, a simple PLA/PETG fiber is fabricated and the co-drawing ability of the selected materials is demonstrated
 - The shape-memory capabilities (shape programming and recovery) of this first fiber are tested
- 2) Pushing further, three shape-memory fibered structures with light-guiding abilities are fabricated based on the first results
 - The shape-memory capabilities of the proposed fibers are tested through shape-recovery cycling experiments
 - Optical performances of the different architectures are also assessed (attenuation measurements)
 - The fiber possessing an ABS cladding and PETG core is performing the best and can withstand bending/straightening cycles
- 3) Finally, a fiber evanescent wave spectroscopy (FEWS) sensor is demonstrated using an exposed-core shape-memory fiber.

- The ability of the waveguide to alter its shape is leveraged to dynamically optimize the performance of the device

We also performed a careful editing of the manuscript to improve the language. For better clarity, we also have reworked and simplified several of the Figures (**see response 1 to reviewer 1**).

REVIEWER COMMENTS

Reviewer #1 (Remarks to the Author):

The authors have addressed all the concerns after this round of revisions.

Reviewer #2 (Remarks to the Author):

I am not satisfied with the authors' responses to my comments. I do not believe this work can meet the criteria of Nature Communications.

Regarding the response to my comment 1, actually many results presented in this work are about the optical function, even with their newly added application based on exposed-core fiber.

Regarding the response to my comment 2, the information given by the initial submission was very misleading. The revised version further weakens the novelty of this work.

Regarding the response to my comment 3, the newly added application is quite confusing, and the performance is poor. What do you want to demonstrate? How can such a sensor be used?

Reviewer #3 (Remarks to the Author):

The author has resolved most of the issues and the revised manuscript can be accepted.

Reviewer #1 (Remarks to the Author):

The authors have addressed all the concerns after this round of revisions.

Reviewer #2 (Remarks to the Author):

I am not satisfied with the authors' responses to my comments. I do not believe this work can meet the criteria of Nature Communications.

1. Regarding the response to my comment 1, actually many results presented in this work are about the optical function, even with their newly added application based on exposed-core fiber.

Although we present applications of shape-memory fibers in the field of optics in the manuscript, another primary contribution of our work is to demonstrate the integration of fiber technology and shape-memory materials. Since the emergence of multi-material fibers, the applications objects fabricated through the highly scalable thermal drawing process have diversified and now cover other areas such as electronics and microfluidics, among others [a, b]. Our results, therefore, also contributes to these research topics.

Additionally, our work contributes to the field of 4D printing and shape-memory objects in general, as **we present shape-changing objects with unprecedented aspect ratios** (length > 50 m and diameter of few hundreds of micrometers) **while still possessing structuration at the micrometric scale**.

The fabricated fibers moreover possess light-guiding abilities, hence their potential for applications in photonics.

This is clearly stated in the final paragraph of the manuscript (page 22-23), where we discuss the applications (for health, robotics, textiles and photonics) of our shape-memory fibers to emphasize the relevance of our research.

[a] Yan, W., Page, A., Nguyen-Dang, T., Qu, Y., Sordo, F., Wei, L., & Sorin, F. (2019). Advanced multimaterial electronic and optoelectronic fibers and textiles. *Advanced materials*, 31(1), 1802348.

[b] Dong, C., Page, A. G., Yan, W., Nguyen-Dang, T., & Sorin, F. (2019). Microstructured multimaterial fibers for microfluidic sensing. *Advanced Materials Technologies*, 4(10), 1900417.

2. Regarding the response to my comment 2, the information given by the initial submission was very misleading. The revised version further weakens the novelty of this work.

We don't agree with the reviewer on this point. Our methodology simply takes advantage of several preform manufacturing techniques, namely 3D printing and a rod-in-tube-like technique, to successfully develop fibered structures with intricate geometries and integrating shape-memory materials while still exhibiting acceptable light transmission. Also, 3D-printing of a waveguiding core with similar attenuation level is still possible as demonstrated in literature where fiber attaining ~0.4 dB/cm losses are reported [a]. More importantly, the fabrication method used for the production of the fiber core does not change the conclusions and perspectives of our work.

We modified the following sentence to clarify this subject (page 11):

“Note that the central piece used for the core is not 3D-printed, which helps in minimizing optical attenuation **although acceptable optical losses of 0.4 dB/cm can be achieved in 3D-printed polymer fiber**¹⁹.”

[a] Luo, Y., Canning, J., Zhang, J., & Peng, G. D. (2020). Toward optical fibre fabrication using 3D printing technology. *Optical Fiber Technology*, 58, 102299. [c] Fu, R., Luo, W., Nazempour, R., Tan, D., Ding, H.,

3. Regarding the response to my comment 3, the newly added application is quite confusing, and the performance is poor. What do you want to demonstrate? How can such a sensor be used?

The results presented in the last section of the manuscript describe proof-of-principle experiments. We demonstrate there the potential of our shape-memory fibers for the development of sensors based on fiber evanescent wave spectroscopy (FEWS).

FEWS sensors rely on the detection of the characteristic absorptions of chemical species of interest. In practice, a broad-spectrum source is propagated through a waveguide directly exposed to an analyte. Interactions between the evanescent field and the analyte causes part of the light to be absorbed during its propagation through the waveguide. Spectral position and amplitude of the detected absorption bands can then be used to identify and quantify the present analyte. The performances, i.e. the sensitivity, of this type of sensor is linked to the evanescent field depth of penetration, which depends on various factors, including the considered spectral range, the angle of attenuated total reflectance (i.e. the curvature radius and the orientation angle of the fiber), as well as the refractive index of the analyte and therefore its concentration.

In the manuscript, we use a shape-memory exposed-core fiber to create an adaptive FEWS sensor. For demonstration purposes, a simple blue dye is used as the analyte. We observe that there exists an optimal orientation angle of the fiber at which the analyte's absorption amplitude is maximized. Maximization of the absorption amplitude is particularly useful to improve the sensitivity of the sensor. This type of adaptive sensor can potentially be used to detect chemical pollutants in aqueous environments for instance. The ability to modify the fiber's orientation angle through shape-memory effects allows for real-time optimization of the sensor's shape based on the considered analyte (its concentration, the spectral position of its absorption bands, etc.).

We added the following sentence in the text (page 22):

“It clearly shows a maximum around 90° which therefore stands out as the optimal angle to perform FEWS measurements in the present configuration, **as maximization of the absorption amplitude is particularly useful to improve the sensitivity of the sensor.**”

Reviewer #3 (Remarks to the Author):

The author has resolved most of the issues and the revised manuscript can be accepted.

REVIEWERS' COMMENTS

Reviewer #2 (Remarks to the Author):

My concerns have been addressed. This submission is ready for publication.